# On Contrastive Representations
# of Stochastic Processes

**Emile Mathieu**[†][*], **Adam Foster**[†][*], **Yee Whye Teh**[†,‡]
{emile.mathieu, adam.foster, y.w.teh}@stats.ox.ac.uk,
† Department of Statistics, University of Oxford, United Kingdom
‡ DeepMind, United Kingdom

## Abstract

Learning representations of stochastic processes is an emerging problem in machine learning with applications from meta-learning to physical object models to time series. Typical methods rely on exact reconstruction of observations, but this approach breaks down as observations become high-dimensional or noise distributions become complex. To address this, we propose a unifying framework for learning contrastive representations of stochastic processes (CRESP) that does away with exact reconstruction. We dissect potential use cases for stochastic process representations, and propose methods that accommodate each. Empirically, we show that our methods are effective for learning representations of periodic functions, 3D objects and dynamical processes. Our methods tolerate noisy high-dimensional observations better than traditional approaches, and the learned representations transfer to a range of downstream tasks.

## 1 Introduction

The stochastic process (Doob, 1953; Parzen, 1999) is a powerful mathematical abstraction used in biology (Bressloff, 2014), chemistry (van Kampen, 1992), physics (Jacobs, 2010), finance (Steele, 2012) and other fields. The simplest incarnation of a stochastic process is a random function $\mathbb{R} \rightarrow \mathbb{R}$, such as a Gaussian Process (MacKay, 2003), that can be used to describe a real-valued signal indexed by time or space. Extending to random functions from $\mathbb{R}$ to another space, stochastic processes can model time-dependent phenomena like queuing (Grimmett and Stirzaker, 2020) and diffusion (Itô et al., 2012). In meta-learning, the stochastic process can be used to describe few-shot learning tasks—mappings from images to class labels (Vinyals et al., 2016)—and image completion tasks—mappings from pixel locations to RGB values (Garnelo et al., 2018a). In computer vision, 2D views of 3D objects can be seen as observations of a stochastic process indexed by the space of possible viewpoints (Eslami et al., 2018; Mildenhall et al., 2020). Videos can be seen as samples from a time-indexed stochastic process with 2D image observations (Zelnik-Manor and Irani, 2001).

Table 1: Example stochastic processes with covariate space $\mathcal{X}$ and observation space $\mathcal{Y}$.

| | $\mathcal{X}$ | $\mathcal{Y}$ | Illustration |
|---|---|---|---|
| 1D function | $\mathbb{R}$ | $\mathbb{R}$ | |
| Image in-fill | $\mathbb{Z}^2$ | $\mathbb{R}^3$ | |
| 3D object | $SE(3)$ | Images | |
| Video | $\mathbb{R}$ | Images | |

---

[*]Equal contribution. Author ordering determined by coin flip.

Machine learning algorithms that operate on data generated from stochastic processes are therefore in high demand. We assume that we have access to only a small set of covariate–observation pairs $\{(\boldsymbol{x}_i, \boldsymbol{y}_i)_{i=1}^C\}$ from different realizations of the underlying stochastic process. This might correspond to a few views of a 3D object, or a few snapshots of a dynamical system evolving in time. Whilst conventional deep learning thrives when there is a large quantity of i.i.d. data available (Lake et al., 2017), allowing us to learn a fresh model for each realization of the stochastic process, when the context size is small it makes sense to use data from other realizations to build up prior knowledge about the domain which can aid learning on new realizations (Reed et al., 2018; Garnelo et al., 2018a).

Traditional methods for learning from stochastic processes, including the Gaussian Process family (MacKay, 2003; Rasmussen, 2003) and the Neural Process family (Garnelo et al., 2018a,b; Eslami et al., 2018), learn to reconstruct a realization of the process from a given context. That is, given a context set $\{(\boldsymbol{x}_i, \boldsymbol{y}_i)_{i=1}^C\}$, these methods provide a predictive distribution $q\left(\boldsymbol{y}^\star | \boldsymbol{x}^\star, (\boldsymbol{x}_i, \boldsymbol{y}_i)_{i=1}^C\right)$ for the observation that would be obtained from this realization of the process at any target covariate $\boldsymbol{x}^\star$. These methods use an explicit likelihood for $q$, typically a Gaussian distribution. Whilst this can work well when $\boldsymbol{y}^\star$ is low-dimensional and unimodal, it is a restrictive assumption. For example, when $p\left(\boldsymbol{y}^\star | \boldsymbol{x}^\star, (\boldsymbol{x}_i, \boldsymbol{y}_i)_{i=1}^C\right)$ samples a high-dimensional image with colour distortion, traditional methods must learn to perform conditional image generation, a notably challenging task (van den Oord et al., 2016; Chrysos and Panagakis, 2021).

In this paper, we do away with the explicit likelihood requirement for learning from stochastic processes. Our first insight is that, for a range of important downstream tasks, exact reconstruction is not necessary to obtain good performance. Indeed, whilst $\boldsymbol{y}$ may be high-dimensional, the downstream target label or feature $\ell$ may be simpler. We consider two distinct settings for $\ell \in L$. The first is a downstream task that *depends on the covariate* $\boldsymbol{x} \in \mathcal{X}$, formally a second process $\mathcal{X} \to L$ that covaries with the first. For example, $\ell(\boldsymbol{x})$ could represent a class label or annotation for each video frame. The second is a downstream task that depends on the entire process realization, such as a single label for a 3D object. In both cases, we assume that we have limited labelled data, so we are in a semi-supervised setting (Zhu, 2005).

To solve problems of this nature, we propose a general framework for Contrastive Representations of Stochastic Processes (CRESP). At its core, CRESP consists of a flexible encoder network architecture for contexts $\{(\boldsymbol{x}_i, \boldsymbol{y}_i)_1^C\}$ that unites transformer encoders of sets (Vaswani et al., 2017; Parmar et al., 2018) with convolutional encoders (LeCun et al., 1989) for observations that are images. To account for the two kinds of downstream task that may of interest, we propose a *targeted* variant of CRESP that learns a representations depending on the context and a target covariate $\boldsymbol{x}^\star$, and an *untargeted* variant that learns one representation of the context. To train our encoder, we take our inspiration from recent advances in contrastive learning (Bachman et al., 2019; Chen et al., 2020) which have so far focused on representations of single observations, typically images. We define a variant of the InfoNCE objective (van den Oord et al., 2018) for contexts sampled from stochastic processes, allowing us to avoid training objectives that necessitate exact reconstruction. Rather than attempting pixel-perfect reconstruction, then, CRESP solves a self-supervised task in representation space.

The CRESP framework unifies and extends recent work, building on function contrastive learning (FCLR) (Gondal et al., 2021) by considering targeted as well as untargeted representations and using self-attention in place of mean-pool aggregation. We develop on noise contrastive meta-learning (Ton et al., 2021) by focusing on downstream tasks rather than multi-modal reconstruction, replacing conditional mean embeddings with neural representations and using a simpler training objective.

We evaluate CRESP on sinusoidal functions, 3D objects, and dynamical processes with high-dimensional observations. We empirically show that our methods can handle high-dimensional observations with naturalistic distortion, unlike explicit likelihood methods, and our representations lead to improved data efficiency compared to supervised learning. CRESP performs well on a range of downstream tasks, both targeted and untargeted, outperforming existing methods across the board. Our code is publicly available at `github.com/ae-foster/cresp`.

## 2 Background

**Stochastic Processes** Stochastic Processes (SPs) are probabilistic objects defined as a family of random variables indexed by a covariate space $\mathcal{X}$. For each $\boldsymbol{x} \in \mathcal{X}$, there is a corresponding random variable $\boldsymbol{y} | \boldsymbol{x} \in \mathcal{Y}$ living in the observation space. For example, $\boldsymbol{x}$ might represent a pose and $\boldsymbol{y}$ a

photograph of an underlying object take from pose $\boldsymbol{x}$ (see Tab. 1). We assume that there is a *realization* $F$ sampled from a prior $p(F)$, and that the random variable $\boldsymbol{y}|\boldsymbol{x}$ is a sample from $p(\boldsymbol{y}|F, \boldsymbol{x})$. Thus, for each $\boldsymbol{x} \in \mathcal{X}$, $F$ defines a conditional distribution $p(\boldsymbol{y}|F, \boldsymbol{x})$. We assume that observations are independent conditional on the realization $F$. Hence, the joint distribution of multiple observations at locations $\boldsymbol{x}_{1:C}$ from one realization of the stochastic process with prior $p(F)$ is

$$p(\boldsymbol{y}_{1:C}|\boldsymbol{x}_{1:C}) = \int p(F) \prod_{i=1}^{C} p(\boldsymbol{y}_i|F, \boldsymbol{x}_i) \, dF. \tag{1}$$

Conversely, assuming *exchangeability* and *consistency*, the Kolmogorov Extension Theorem guarantees that the joint distribution takes the form (1) (Øksendal, 2003; Garnelo et al., 2018b).

**Neural Processes**  The neural process (NP) and conditional neural process (CNP) are closely related models that learn representations of data generated by a stochastic process (SP)[1]. The training objective for the NP and CNP is inspired by the posterior predictive distribution for SPs: given a context $\{(\boldsymbol{x}_i, \boldsymbol{y}_i)_{i=1}^{C}\}$, the observation at the target covariate $\boldsymbol{x}^\star$ has the distribution

$$p(\boldsymbol{y}^\star|\boldsymbol{x}^\star, (\boldsymbol{x}_i, \boldsymbol{y}_i)_{i=1}^{C}) = \int p(F|(\boldsymbol{x}_i, \boldsymbol{y}_i)_{i=1}^{C}) \, p(\boldsymbol{y}^\star|F, \boldsymbol{x}^\star) \, dF. \tag{2}$$

The CNP learns a neural approximation $q\left(\boldsymbol{y}^\star|\boldsymbol{x}^\star, (\boldsymbol{x}_i, \boldsymbol{y}_i)_{i=1}^{C}\right) = p(\boldsymbol{y}^\star|\boldsymbol{c}, \boldsymbol{x}^\star)$ to equation (2), where $\boldsymbol{c} = \sum_i g_{\mathrm{enc}}(\boldsymbol{x}_i, \boldsymbol{y}_i)$ is a permutation-invariant *context representation* and $p(\cdot|\boldsymbol{c}, \boldsymbol{x})$ is an explicit likelihood. Conventionally, $p$ is a Gaussian with mean and variance given by a neural network applied to $\boldsymbol{c}, \boldsymbol{x}$. The CNP model is then trained by maximum likelihood. In the NP model, an additional latent variable $\boldsymbol{u}$ is used to represent uncertainty in the process realization, more closely mimicking (2).

A significant limitation, common to the NP family, is the reliance on an explicit likelihood. Indeed, requiring $\log q\left(\boldsymbol{y}^\star|\boldsymbol{x}^\star, (\boldsymbol{x}_i, \boldsymbol{y}_i)_{i=1}^{C}\right)$ to be large requires the model to successfully reconstruct $\boldsymbol{y}^\star$ based on the context, similarly to the reconstruction term in variational autoencoders (Kingma and Welling, 2014). Furthermore, the NP objective cannot be increased by extracting additional features from the context unless the predictive part of the model, the part mapping from $(\boldsymbol{c}, \boldsymbol{x})$ to a mean and variance, is powerful enough to use them.

**Contrastive Learning and Likelihood-free Inference**  Contrastive learning has enjoyed recent success in learning representations of high-dimensional data (van den Oord et al., 2018; Bachman et al., 2019; He et al., 2020; Chen et al., 2020), and is deeply connected to likelihood-free inference (Gutmann and Hyvärinen, 2010; van den Oord et al., 2018; Durkan et al., 2020). In its simplest form, suppose we have a distribution $p(\boldsymbol{y}, \boldsymbol{y}')$, for example $\boldsymbol{y}$ and $\boldsymbol{y}'$ could be differently augmented versions of the same image. Rather than fitting a model to predict $\boldsymbol{y}'$ given $\boldsymbol{y}$, which would necessitate high-dimensional reconstruction, contrastive learning methods can be seen as learning the likelihood-ratio $r(\boldsymbol{y}'|\boldsymbol{y}) = p(\boldsymbol{y}'|\boldsymbol{y})/p(\boldsymbol{y}')$. To achieve this, contrastive methods encode $\boldsymbol{y}, \boldsymbol{y}'$ to deterministic embeddings $\boldsymbol{z}, \boldsymbol{z}'$, and consider additional 'negative' samples $\boldsymbol{z}'_1, ..., \boldsymbol{z}'_{K-1}$ which are the embeddings of other independent samples of $p(\boldsymbol{y}')$ (for example, taken from the same training batch as $\boldsymbol{y}, \boldsymbol{y}'$). The InfoNCE training loss (van den Oord et al., 2018) is then given by

$$\mathcal{L}_K^{\mathrm{InfoNCE}} = -\mathbb{E}\left[\log \frac{s(\boldsymbol{z}, \boldsymbol{z}')}{s(\boldsymbol{z}, \boldsymbol{z}') + \sum_k s(\boldsymbol{z}, \boldsymbol{z}'_k)}\right] - \log K. \tag{3}$$

for similarity score $s > 0$. Informally, InfoNCE is minimized when $\boldsymbol{z}$ is more similar to $\boldsymbol{z}'$—the 'positive' sample—than it is to the negative samples $\boldsymbol{z}'_1, ..., \boldsymbol{z}'_{K-1}$ that are independent of $\boldsymbol{z}$. Formally, Eq. (3) is the multi-class cross-entropy loss arising from classifying the positive sample correctly. It can be shown that the optimal similarity score $s$ is proportional to the true likelihood ratio $r$ (van den Oord et al., 2018; Durkan et al., 2020). A key feature of InfoNCE is that learns about the predictive density $p(\boldsymbol{y}'|\boldsymbol{y})$ indirectly, rather than by attempting direct reconstruction.

## 3  Method

Given data $\{(\boldsymbol{x}_i, \boldsymbol{y}_i)_{i=1}^{C}\}$ sampled from a realization of a stochastic process, one potential task is to make predictions about how observations will look at another $\boldsymbol{x}^\star$—this is the task that is solved

---

[1]Note that neither the neural process (NP) nor the conditional neural process (CNP) is formally stochastic processs (SPs) as they do *not* satisfy the consistency property.

by the NP family. However, in practice the inference that we want to make from the context data could be different. For instance, rather than predicting a high-dimensional observation at a future time or another location, we could be interested in inferring some low-dimensional feature of that observation—whether two objects have collided at that point in time, or if an object can be seen from a given pose. Even more simply, we might be solely interested in classifying the context, deciding what object is being viewed, for example. Such *downstream tasks* provide a justification for learning representations of stochastic processes that are not designed to facilitate predictive reconstruction of the process at some $\boldsymbol{x}^\star$. We break downstream tasks for stochastic processes into two categories.

**Targeted and untargeted tasks** A *targeted* task is one in which the label $\ell$ depends on $\boldsymbol{x}$, as well as on the underlying realization of the process $F$. This means that we augment the stochastic process of Sec. 2 by introducing a conditional distribution $p(\ell|F,\boldsymbol{x})$. The goal is to infer the predictive density $p\left(\ell^\star|\boldsymbol{x}^\star,(\boldsymbol{x}_i,\boldsymbol{y}_i)_{i=1}^C\right)$. An *untargeted task* associates one label $y$ with the entire realization $F$ via a conditional distribution $p(\ell|F)$. The aim is to infer the conditional distribution $p\left(\ell|(\boldsymbol{x}_i,\boldsymbol{y}_i)_{i=1}^C\right)$.

**Representation learning** We assume a semi-supervised (Zhu, 2005) setting, with unlabelled contexts for a large number of realizations of the stochastic process, but few labelled realizations. To make best use of this unlabelled data, we learn representations of contexts, and then fit a downstream model on top of fixed representations. In the stochastic process context, we have the requirement for a representation learning approach that can transfer to both targeted and untargeted downstream tasks. We therefore propose a general framework to learn contrastive representations of stochastic processes (CRESP). Our framework consists of a flexible encoder architecture that processes the context $\{(\boldsymbol{x}_i,\boldsymbol{y}_i)_{i=1}^C\}$ and a $\boldsymbol{x}^\star$-dependent head for targeted tasks. This means CRESP can encode data from stochastic processes in two ways: 1) a *targeted* representation that depends on the context $\{(\boldsymbol{x}_i,\boldsymbol{y}_i)_{i=1}^C\}$ and some target location $\boldsymbol{x}^\star$, being a predictive representation for the process at this covariate, suitable for targeted downstream tasks; or 2) a single *untargeted* representation of the context $\{(\boldsymbol{x}_i,\boldsymbol{y}_i)_{i=1}^C\}$ that summarizes the entire realization $F$, suitable for untargeted tasks.

## 3.1 Training

We have unlabelled data $\{(\boldsymbol{x}_i,\boldsymbol{y}_i)_{i=1}^C\}$ that is generated from the stochastic process (1), but unlike the Neural Process family, we do not wish to place an explicit likelihood on the observation space $\mathcal{Y}$. Instead, we adopt a contrastive self-supervised learning approach (van den Oord et al., 2018; Bachman et al., 2019; Chen et al., 2020) to training. Whilst we adopt subtly different training schemes for the targeted and untargeted cases, the broad strokes are the same. Given a mini-batch of contexts samples from different realizations of the underlying stochastic process, create predictive and ground truth representations from each. We then use representations from other observations in the same mini-batch as negative samples in an InfoNCE-style (van den Oord et al., 2018) training loss. This can be seen as learning an *unnormalized* likelihood ratio. Taking gradients through this loss function allows us to update our CRESP network by gradient descent (Robbins and Monro, 1951). We now describe the key differences between the *targeted* and *untargeted* cases.

**Targeted CReSP** This setting is closer in spirit to the CNP. Rather than making a direct estimate of the posterior predictive $p(\boldsymbol{y}^\star|\boldsymbol{x}^\star,(\boldsymbol{x}_i,\boldsymbol{y}_i)_{i=1}^C)$ for each value of $\boldsymbol{x}^\star$, we instead attempt to learn the following likelihood-ratio

$$r(\boldsymbol{y}^\star|\boldsymbol{x}^\star,(\boldsymbol{x}_i,\boldsymbol{y}_i)_{i=1}^C) = \frac{p(\boldsymbol{y}^\star|\boldsymbol{x}^\star,(\boldsymbol{x}_i,\boldsymbol{y}_i)_{i=1}^C)}{p(\boldsymbol{y}^\star)} \tag{4}$$

where $p(\boldsymbol{y}^\star)$ is the marginal distribution of observations from different realizations of the process and different covariates. To estimate this ratio with contrastive learning, we first randomly separate the context $\{(\boldsymbol{x}_i,\boldsymbol{y}_i)_{i=1}^C\}$ into a training context $\{(\boldsymbol{x}_i,\boldsymbol{y}_i)_{i=1}^{C-1}\}$ and a target $(\boldsymbol{x}^\star,\boldsymbol{y}^\star)$. We then process $\{(\boldsymbol{x}_i,\boldsymbol{y}_i)_{i=1}^{C-1},\boldsymbol{x}^\star\}$ and $\boldsymbol{y}^\star$ separately with an encoder network, yielding respectively a predictive representation $\hat{\boldsymbol{c}}$ and a target representation $\boldsymbol{c}^\star$. This encoder network is described in detail in the following Sec. 3.2. These representations are further projected into a low-dimensional space $\mathcal{Z}$ using a shallow MLP, referred as `Projection head` on Fig. 1, giving $\hat{\boldsymbol{z}}$ and $\boldsymbol{z}^\star$. We create negative samples $\boldsymbol{z}_1',...,\boldsymbol{z}_{K-1}'$, defined as samples coming from other realisations of the stochastic process, from representations obtained from the other observations of the batch. This means that we are

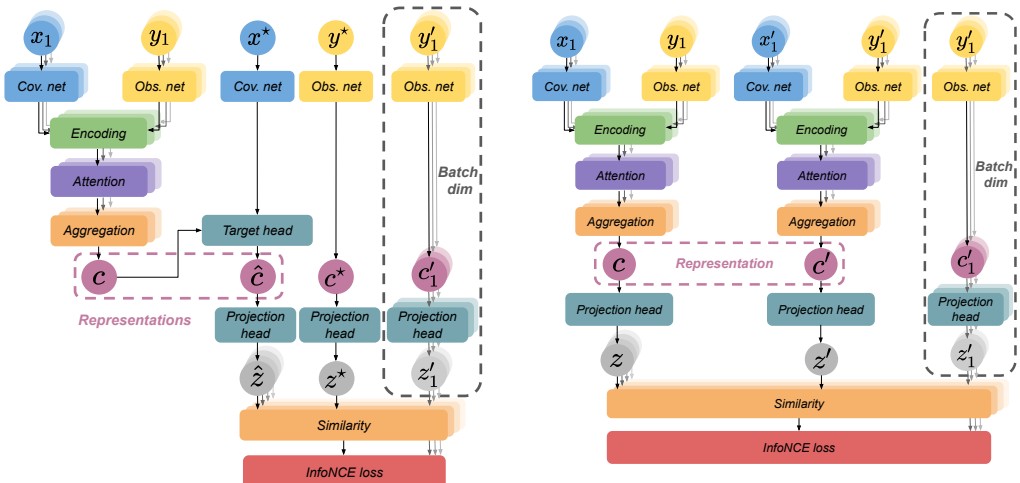

Figure 1: CReSP architecture with contrastive loss. [Left] Targeted, [Right] Untargeted.

drawing negative samples via the distribution $p(\boldsymbol{y}^\star)$ as required for (4). We then form the contrastive loss

$$\mathcal{L}_K^{\text{targeted}} = -\mathbb{E}\left[\log \frac{s(\boldsymbol{z}^\star, \hat{\boldsymbol{z}})}{s(\boldsymbol{z}^\star, \hat{\boldsymbol{z}}) + \sum_k s(\boldsymbol{z}'_k, \hat{\boldsymbol{z}})}\right] - \log K \tag{5}$$

with $s(\boldsymbol{z}^\star, \hat{\boldsymbol{z}}) = \exp\left(\boldsymbol{z}^{\star\top}\hat{\boldsymbol{z}}/\tau\|\boldsymbol{z}^\star\|\|\hat{\boldsymbol{z}}\|\right)$. By minimizing this loss, we ensure that the predicted representation is closer to the representation of the true outcome than representations of other random outcomes. The optimal value of $s(\boldsymbol{z}^\star, \hat{\boldsymbol{z}})$ is proportional to the likelihood ratio $r(\boldsymbol{y}^\star|\boldsymbol{x}^\star, (\boldsymbol{x}_i, \boldsymbol{y}_i)_{i=1}^C)$.

**Untargeted CReSP**  For the untargeted version, we simply require a representation of each context; $\boldsymbol{x}^\star$ no longer plays a role. The key idea here is that, without estimating a likelihood ratio in $\mathcal{Y}$ space, we can use contrastive methods to encourage two representations formed from the *same realization* of the stochastic process to be more similar than representations formed from *different* realizations. To achieve this, we randomly split the whole context $\{(\boldsymbol{x}_i, \boldsymbol{y}_i)_{i=1}^C\}$ into two training contexts $\{(\boldsymbol{x}_i, \boldsymbol{y}_i)_{i=1}^{C_1}\}$ and $\{(\boldsymbol{x}'_i, \boldsymbol{y}'_i)_{i=1}^{C_2}\}$, with an equal split $C_1 = C_2 = C/2$ being our standard approach. We encode both with an encoder network, giving two representations $\boldsymbol{c}, \boldsymbol{c}'$, further projected into lower-dimensional representations $\boldsymbol{z}, \boldsymbol{z}'$ as in the targeted case. We also take $K - 1$ negative samples $\boldsymbol{z}'_1, ..., \boldsymbol{z}'_K$ using other representations in the same training mini-batch.

$$\mathcal{L}_K^{\text{untargeted}} = -\mathbb{E}\left[\log \frac{s(\boldsymbol{z}, \boldsymbol{z}')}{s(\boldsymbol{z}, \boldsymbol{z}') + \sum_k s(\boldsymbol{z}, \boldsymbol{z}'_k)}\right] - \log K. \tag{6}$$

This training method is closer in spirit to SimCLR (Chen et al., 2020), but here we include attention and aggregation steps to combine the distinct elements of the context.

### 3.2 Representation

The core of our architecture is a flexible encoder of a context $\{(\boldsymbol{x}_i, \boldsymbol{y}_i)_{i=1}^C\}$, as illustrated in Fig. 1.

**Covariate and observation preprocessing**  We begin by applying separate networks to the covariate $g_{\text{cov}}(\boldsymbol{x})$ and observation $g_{\text{obs}}(\boldsymbol{y})$ of each pair $(\boldsymbol{x}, \boldsymbol{y})$ of the context. When observations $\boldsymbol{y}$ are high-dimensional, such as images, this step is crucial because we can use existing well-developed vision architectures such as CNNs (LeCun et al., 1989) and ResNets (He et al., 2016) to extract image features. For covariates that are angles, we use Random Fourier Features (Rahimi and Recht, 2008).

**Pair encoding**  We then combine separate encodings of $\boldsymbol{x}, \boldsymbol{y}$ into a single representation for the pair. We concatenate the individual representations and pass them through a simple neural network, i.e. $g_{\text{enc}}(\boldsymbol{x}, \boldsymbol{y}) := g_{\text{enc}}([g_{\text{cov}}(\boldsymbol{x}), g_{\text{obs}}(\boldsymbol{y})])$. In practice, we found that a gated architecture works well.

**Attention & Aggregation** We apply self-attention (Vaswani et al., 2017) over the $C$ different encodings of the context $\{g_{\text{enc}}(\boldsymbol{x}_i, \boldsymbol{y}_i)\}_{i=1}^C$. We found transformer attention (Parmar et al., 2018) to perform best. We then pool the $C$ reweighted representations to yield a single representation $\boldsymbol{c} = \sum_i g_{\text{enc}}(\boldsymbol{x}_i, \boldsymbol{y}_i)$. For targeted representations, we concatenate $\boldsymbol{c}$ and $\boldsymbol{x}^\star$, then pass them through a *target head* yielding $\hat{\boldsymbol{c}} = h(\boldsymbol{x}^\star, \boldsymbol{c})$, the predictive representation at $\boldsymbol{x}^\star$.

### 3.3 Transfer to downstream tasks

We have outlined the *unsupervised* part of CRESP—a way to learn a representation of a context sampled from a stochastic process without explicit reconstruction. We now return to our core motivation for such representations, which is to use them to solve a downstream task, either targeted or untargeted. This will be particularly useful in a *semi-supervised* setting, in which labelled data for the downstream task is limited compared to the unlabelled data used for unsupervised training of the CRESP encoder. Our general approach to both targeted and untargeted downstream tasks is to fit *linear* models on the context representations of the labelled training set, and use these to predict labels on new, unseen realizations of the stochastic process, following the precedent in contrastive learning (Hjelm et al., 2019; Kolesnikov et al., 2019). We do not use fine-tuning.

For targeted tasks, we assume that we have labelled data from $n$ realizations of the stochastic process that takes the form of an *unlabelled* context $(\boldsymbol{x}_{ij}, \boldsymbol{y}_{ij})_{i=1}^C$ along with a *labelled pair* $(\boldsymbol{x}_j^\star, \ell_j^\star)$ for each $j = 1, \ldots, n$. Here, $\ell_j^\star$ is the label at location $\boldsymbol{x}^\star$ for realization $j$. To fit a downstream classifier using CRESP representations with this labelled dataset, we first process each $(\boldsymbol{x}_{ij}, \boldsymbol{y}_{ij})_{i=1}^C$ along with the covariate $\boldsymbol{x}_j^\star$ through a *targeted* CRESP encoder to produce $\hat{\boldsymbol{c}}_j$. This allows us to form a training dataset $(\hat{\boldsymbol{c}}_j, \ell_j^\star)_{j=1}^n$ of representation, label pairs which we then use to train our downstream classifier. At test time, given a test context $(\boldsymbol{x}_i', \boldsymbol{y}_i')_{i=1}^C$, we can predict the unknown label at any $\boldsymbol{x}^\star$ by forming the corresponding targeted representation with the CRESP network, and then feeding this into the linear classifier. This is akin to zero-shot learning (Xian et al., 2018).

For untargeted tasks, the downstream model is simpler. Given labelled data consisting of contexts $(\boldsymbol{x}_{ij}, \boldsymbol{y}_{ij})_{i=1}^C$ with label $\ell_j$ for $j = 1, \ldots, n$, we can use the untargeted CRESP encoder to produce a training dataset $(\boldsymbol{c}_j, \ell_j)$ as before. Actually, *targeted* CRESP can also be used to obtain *untargeted* representations $\boldsymbol{c}_j$— without applying the target head. We then use this to train the linear classifier. At test time, we predict labels for contexts from new, unseen realizations of the stochastic process.

## 4   Related work

**Neural process family** Neural Processes (Garnelo et al., 2018b) and Conditional Neural Processes (Eslami et al., 2018; Garnelo et al., 2018a) are closely related methods that create a representation of an stochastic process realization by aggregating representations of a context. Unlike CRESP, NPs are generative models that uses an explicit likelihood, generally a fully factorized Gaussian, to estimate the posterior predictive distribution. Attentive (Conditional) Neural Processes (Kim et al., 2019, A(C)NP) introduced both self-attention and cross-attention into the NP family. The primary distinction between this family and CRESP is the explicit likelihood that is used for reconstruction. As the most comparable method to CRESP, we focus on the (A)CNP in the experiments.

**SimCLR family** Recent popular methods in contrastive learning (van den Oord et al., 2018; Bachman et al., 2019; Tian et al., 2020; Chen et al., 2020) create neural representations of single objects, typically images, that are approximately invariant to a range of transformations such as random colour distortion. Like CRESP, many of these approaches use the InfoNCE objective to train encoders. What distinguishes CRESP from conventional contrastive learning methods is that it provides representations of realizations of stochastic processes, rather than of individual images. Thus, standard contrastive learning solves a strictly less general problem than CRESP in which the covariate $\boldsymbol{x}$ is absent. Standard contrastive encoders do not aggregate multiple covariate-observation pairs of a context, although simpler feature averaging (Foster et al., 2020) has been applied successfully.

**Function contrastive learning** In their recent paper, Gondal et al. (2021) considered function contrastive learning (FCLR) which uses a self-supervised objective to learn representations of functions. FCLR fits naturally into the CRESP framework as an *untargeted* approach that uses

mean-pooling in place of our attention aggregation. Conceptually, then, FCLR does not take account of targeted tasks, nor does it propose a method for targeted representation learning.

**Noise contrastive meta-learning**    Ton et al. (2021) proposed an approach for conditional density estimation in meta-learning, motivated by multi-modal reconstruction. Like targeted CRESP, their method targets the unnormalized likelihood ratio (4). They use a noise contrastive (Gutmann and Hyvärinen, 2010) training objective with an explicitly defined 'fake' distribution that is different from the CRESP training objective. Their primary method, MetaCDE, uses conditional mean embeddings to aggregate representations, unlike our attentive aggregation. This means that, when using it as a baseline within our framework, MetaCDE does not form a fixed-dimensional representation of contexts, and so cannot be applied to untargeted tasks. They also proposed MetaNN, a purely neural version of their main approach.

## 5    Experiments

We consider three different stochastic processes and downstream tasks which possess high-dimensional observations or complex noise distributions: 1) inferring parameters of periodic functions, 2) classifying 3D objects and 3) predicting collisions in a dynamical process. We compare several models summarized in Tab. 2 to learn representations of these stochastic processes. All models share the same core encoder architecture. Please refer to Appendix D for full experimental details.

Table 2: Comparison of models used in at least one experiment in Sec. 5.

| Criteria | CNP | ACNP | FCLR | MetaCDE | Targeted CRESP | Untargeted CRESP |
|---|---|---|---|---|---|---|
| **Targeted** | No | No | No | Yes | Yes | No |
| **Reconstruction** | Yes | Yes | No | No | No | No |
| **Attention** | No | Yes | No | No | Yes | Yes |

### 5.1    Sinusoids

We first aim to demonstrate that reconstruction-based methods like CNPs cannot cope well with a bi-modal noise process since their Gaussian likelihood assumption renders them misspecified. We focus on a synthetic dataset of sinusoidal functions with both the observations and the covariates living in $\mathbb{R}$, i.e. $\mathcal{X} = \mathbb{R}$ and $\mathcal{Y} = \mathbb{R}$. We sample one dimensional functions $F \sim p(F)$ such that $F(x) = \alpha \sin(2\pi/T \cdot x + \varphi)$ with random amplitude $\alpha \sim \mathcal{U}([0.5, 2.0])$, phase $\varphi \sim \mathcal{U}([0, \pi])$ and period $T = 8$. We break the uni-modality by assuming a bi-modal likelihood: $p(y|F, x) = 0.5\,\delta_{F(x)}(y) + 0.5\,\delta_{F(x)+\sigma}(y)$ (see Fig. 2a). Context points $x \in \mathcal{X}$ are uniformly sampled in $[-5, 5]$. We consider the *untargeted* downstream task of recovering the functions parameters $\ell = \{\alpha, \varphi\}$, and consequently put to the test our *untargeted* CRESP model along with FCLR and ACNP. We train all models for 200 epochs, varying the distance between modes and the number of training context points. We observe from Fig. 2b that for high intermodal distance, the ACNP is unable to accurately

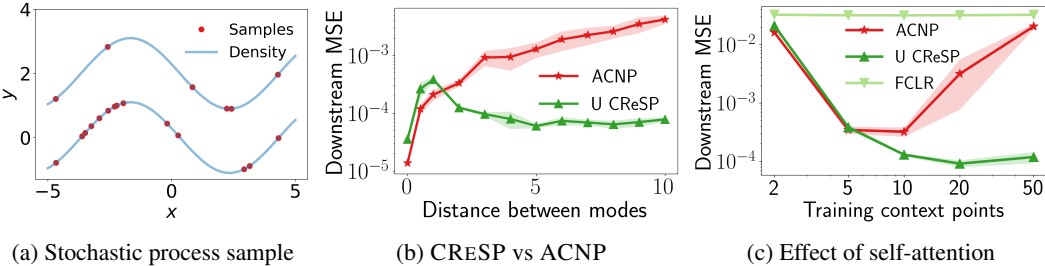

(a) Stochastic process sample          (b) CRESP vs ACNP          (c) Effect of self-attention

Figure 2: We use CRESP along with ACNP and FCLR to recover sinusoid parameters with a bi-modal likelihood. In each setting, we used 20 test views to form representations of the entire training set and fitted a linear classifier to predict the function parameters. Encoders and decoder are MLPs. (a) Visualization of conditional likelihood $p(x|F, \boldsymbol{x})$. (b)(c) Shaded areas represent 95% confidence interval calculated using 6 separately trained networks. We use the shorthand U = untargeted. In (b) we used 10 training views and in (c) the distance between the modes is set to 2.

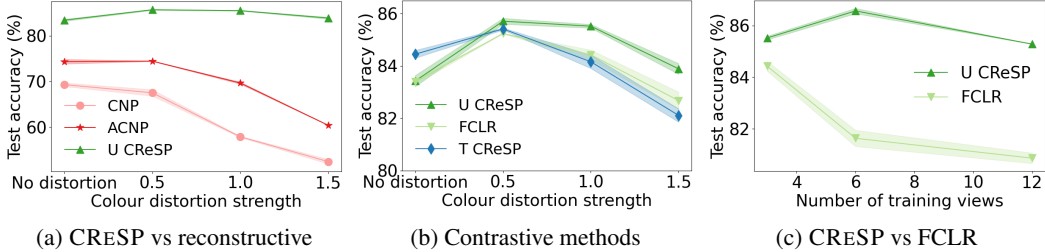

(a) CRESP vs reconstructive     (b) Contrastive methods     (c) CRESP vs FCLR

Figure 4: We compare CRESP with various baseline methods. In each case, we use 10 test views to form representations of the entire training set and fitted a linear classifier to predict ShapeNet object labels. Encoder networks were lightweight CNNs. In (a)(b) we used 3 training views, in (c) we used distortion strength 1. We present the test accuracy $\pm 1$ s.e. and we use the shorthand U = untargeted, T = targeted in figure legends.

recover the true parameters as opposed to CRESP, which is more robust to this bi-modal noise even for distant modes. Additionally, we see in Fig. 2c that self-attention is crucial to accurately recover the sinusoids parameters, as the MSE is several order of magnitude lower for CRESP than for FCLR. We also see that CRESP is able to utilize a larger context better than ACNP.

## 5.2 ShapeNet

We apply CRESP to ShapeNet (Chang et al., 2015), a standard dataset in the field of 3D object representations. Each 3D object can be seen as a realization of a stochastic process with covariates $x$ representing viewpoints. We sample random viewpoints involving both orientation and proximity to the object, with observations $y$ being $64 \times 64$ images taken from that point. We also apply randomized colour distortion as a noise process on the 2D images (see Fig. 3). As the likelihood of this noise process is not known in closed from, this should present a particular challenge to explicit likelihood driven models. The downstream task for ShapeNet is a 13-way object classification which associates a single label with each realization—an untargeted task.

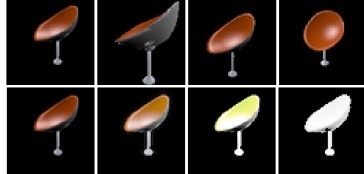

Figure 3: The ShapeNet dataset can be seen as a stochastic process: the covariate $x$ is the viewpoint and the observation $y$ is an image of the object from that viewpoint. [Top] We illustrate an object viewed from 4 random viewpoints. [Bottom] We show varying strengths of colour distortion applied to the same observation, the lefthand column is no distortion.

**CRESP outperforms reconstructive models**    Since the CNP learns by exact reconstruction of observations, we would expect it to struggle with high-dimensional image observations, and particularly suffer as we introduce colour distortion, which is a highly non-Gaussian noise process. To verify this, we trained CNP and ACNP models, along with an attentive untargeted CRESP model which we would expect to perform well on this task. We used the same CNN observation processing network for each method, and an additional CNN decoder for the CNP and ACNP. Fig. 4a shows that CRESP significantly outperforms both the CNP and ACNP, with reconstructive methods faring worse as the level of colour distortion is increased; CRESP actually benefits from mild distortion.

**CRESP outperforms previous contrastive methods**    We next compare different contrastive approachs along two axes: targeted vs untargeted, and attentive vs pool aggregation. This allows a comparison with FCLR (Gondal et al., 2021), which is an untargeted pool-based method. Fig. 4b shows that no contrastive approach performs as badly as the reconstructive methods. Untargeted CRESP performs best, while the targeted method does less well on this untargeted downstream task. With our CNN encoders and a matched architecture for a fair comparison, FCLR does about as well as attentive targeted CRESP and worse than the untargeted counterpart. To further examine the benefits of the attention mechanism used in CRESP, we vary the number of views used during training, focusing on untargeted methods. Fig. 4c shows that as we increase the number of training views, the attentive method outperforms the non-attentive FCLR by an increasing margin. This indicates that careful aggregation and weighting of different views of each object is essential for learning the best representations. The degradation in the performance of FCLR as more training

views are used is likely due to a weaker training signal for the encoder as the self-supervised task becomes easier, this phenomenon also explains why CRESP slightly *decreases* in performance from 6 to 12 training views.

**CRESP benefits from improved label efficiency** We compare CRESP with semi-supervised learning that does not use any pre-training, but instead trains the entire architecture on the labelled dataset. In Fig. 5a we see that pre-training with CRESP can outperform supervised learning on the same fixed dataset at *every label fraction including 100%*. Another axis of variation in the stochastic process setting is the number $C$ of views aggregated at test time. In Fig. 5b, we see that performance increases across the board as we make more views available to form test representations, but that CRESP performs best in all cases.

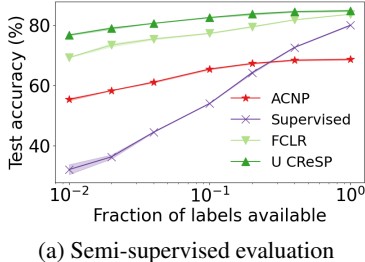
(a) Semi-supervised evaluation

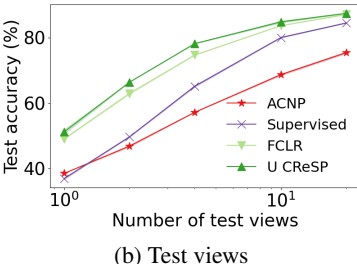
(b) Test views

Figure 5: CRESP for semi-supervised learning. We re-trained the final linear classifiers with different quantities of labelled data and number of test views, supervised learning trained the entire encoder architecture on the same labelled datasets. (a) We used 10 test views, (b) We used 100% of labels. Other settings were as in Fig. 4.

## 5.3 Snooker dynamical process over images

We now focus on the setting where downstream tasks *depend on the covariate* $x^\star$, i.e. *targeted* downstream tasks. In particular, we consider a dynamical system that renders 2D images of two objects with constant velocities and evolving through time as illustrated in Fig. 6a. The objects are constrained in a $1 \times 1$ box and collisions are assumed to result in a perfect reflection. The observation space $\mathcal{Y}$ is consequently the space of $28 \times 28$ RGB images, whilst the covariate space is $\mathbb{R}$, representing time. We consider the downstream task of predicting whether the two objects are overlapping at a given time $x^\star = t$ or not. This experiment aims to reproduce, in a stripped-down manner, the real world problem of collision detection. Even though the object's position can be expressed in closed-form, it is non trivial to predict the 2D image at a specific time given a collection of snapshots. We expect *targeted* CRESP to be particularly well-suited for such a task since the model is learning to form and match a targeted representation to the representation of the ground truth observation thorough the *unsupervised* task.

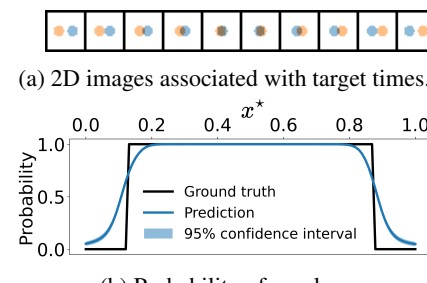
(a) 2D images associated with target times.

(b) Probability of overlap.

Figure 6: We assess the capacity of targeted CRESP to smoothly predict whether the objects are overlapping at a given time $x^\star$ given a context set of size 5. [Top] 2D images associated to $t \in [0, 1]$. [Bottom] Confidence interval is computed over 50 random contexts and 6 trained models.

**CRESP outperforms reconstructive and previous contrastive methods** Alongside targeted CRESP, we consider the CNP, FCLR and MetaCDE models. They are trained for 200 epochs, with contexts of 5 randomly sampled pairs $\{\boldsymbol{y}_i = F(\boldsymbol{x}_i), \boldsymbol{x}_i \sim \mathcal{U}([0, 1])\}$. The encoder is a ResNet18 (He et al., 2016). We found that self-attention did not seem to help any method for this task, so we report un-attentive models. Both CNP and FCLR learn *untargeted* representations during the unsupervised task. We thus feed the downstream linear classifier with the concatenation $\{\boldsymbol{c}, \boldsymbol{x}^\star\}$. Conversely, targeted CRESP and MetaCDE directly produce a *targeted* representation $\hat{\boldsymbol{c}} = h(\boldsymbol{x}^\star, \boldsymbol{c})$ (see Sec. 3.1). The downstream classifier can simply rely on $\hat{\boldsymbol{c}}$ to predict the overlap label $\ell^\star$. We

consequently expect such a targeted representation $\hat{c}$ to be better correlated with the downstream label than untargeted representations $c$.

We observe from Tab. 3 that targeted CRESP significantly outperforms both likelihood-based and previous contrastive methods, though MetaCDE outperforms both untargeted methods (CNP and FCLR). This highlights the need for the *targeted* contrastive loss from Eq. (5) along with a flexible *target head* $h$ to learn targeted representations. Additionally, we observe that in the absence of a noise process, CNP performs as well as FCLR. We further investigate the quality of the learned targeted representations. To do so, given a fixed context we make an overlap prediction at different points in time as shown in Fig. 6b. We observe that targeted CRESP has successfully learned to smoothly predict the overlap label, but also to be uncertain when the overlap is ambiguous. Thus CRESP can successfully interpolate and extrapolate the semantic feature of interest (overlap) without reconstruction.

Table 3: We examine how well learned representations can predict whether the two snooker balls overlap at randomly sampled test times. $95\%$ confidence intervals were computed over 6 runs.

|  | CNP | FCLR | Targeted CRESP | MetaCDE |
|---|---|---|---|---|
| **Accuracy (%)** | $85.3_{\pm 0.5}$ | $85.6_{\pm 0.3}$ | $\mathbf{96.8}_{\pm 0.1}$ | $87.7_{\pm 0.3}$ |

## 6 Discussion

**Limitations** Our method directly learns representations from stochastic processes, without performing reconstruction on the observations, thus if one requires prediction in the observation space $\mathcal{Y}$ then our method cannot be directly applied. Whilst our method is tailor made for a setting of limited *labelled* data, we require access to a large quantity of *unlabelled* data to train our encoder network.

In this work, we do not place uncertainty over context representations. Learning stochastic embeddings would have the primary benefit of producing correlated predictions at two or more covariates, similarly to NPs. As there is no trivial nor unique way to extend the InfoNCE loss to deal with distributions (e.g. Wu and Goodman, 2020), we leave such an extension of our method to future work.

**Future applications** One potential use of CRESP is to generate representations that can be used for reinforcement learning, following the approach of Eslami et al. (2018). One of the key differences between real environments and toy environments is the presence of high-dimensional observations with naturalistic noise. This is a case where the contrastive approach can bring an edge because naturalistic noise significantly damages explicit likelihood methods, but CRESP continues to perform well with more distortion.

**Conclusion** In this work, we introduced a framework for learning contrastive representation of stochastic processes (CRESP). We proposed two variants of our method specifically designed to effectively tackle *targeted* and *untargeted* downstream tasks. By doing away with exact reconstruction, CRESP directly works in the representation space, bypassing any challenge due to high dimensional and multimodal data reconstruction. We empirically demonstrated that our methods are effective for dealing with multi-modal and naturalistic noise processes, and outperform previous contrastive methods for this domain on a range of downstream tasks.

## Acknowledgments

We would like to thank Yann Dubois and Jef Ton for valuable discussions. We also thank Hyunjik Kim, Neil Band and Lewis Smith for providing feedback on earlier versions of the paper. EM research leading to these results received funding from the European Research Council under the European Union's Seventh Framework Programme (FP7/2007- 2013) ERC grant agreement no. 617071 and he acknowledges Microsoft Research and EPSRC for funding EM's studentship. AF gratefully acknowledges funding from EPSRC grant no. EP/N509711/1.

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
