# Appendix for

# On Contrastive Representations
# of Stochastic Processes

## Appendix A    Broader impact

The work presented in this paper focuses on the learning of representations for stochastic processes. Applications in the field of computer vision could lead to better understanding of 3D scenes. Such applications could in turns lead to improved safety in products such as self-driving cars, as well as improved performance in areas such as medical imaging. Nonetheless, as with any computer vision technique, it might also be used in a way that carries societal risk. As a foundational method, our work inherits the broader ethical aspects and future societal consequences of machine learning in general.

## Appendix B    Additional background

**Neural Processes**    Neural processes (NPs) learn a neural approximation $q\left(\boldsymbol{y}^{\star}|\boldsymbol{x}^{\star},(\boldsymbol{x}_i,\boldsymbol{y}_i)_{i=1}^{C}\right)$ to the posterior predictive distributions for stochastic processes given in Eq. (2). To create an efficient neural network architecture, the NP family use the fact that the posterior predictive distribution is unchanged under a permutation of the order $1,...,C$ of the context points. The CNP combines representations of the observed data $\boldsymbol{x}_{1:C},\boldsymbol{y}_{1:C}$ into a *context representation* $\boldsymbol{c}$. To respect the permutation-invariance property, the CNP representation is of the form $\boldsymbol{c}=\sum_c g_{\text{enc}}(\boldsymbol{x}_c,\boldsymbol{y}_c)$ where $g_{\text{enc}}:\mathcal{X}\times\mathcal{Y}\to\mathcal{C}$ is an encoder. The CNP predictions are then given by

$$q\left(\boldsymbol{y}^{\star}|\boldsymbol{x}^{\star},(\boldsymbol{x}_i,\boldsymbol{y}_i)_{i=1}^{C}\right)=p_\theta(\boldsymbol{y}^{\star}|\boldsymbol{c},\boldsymbol{x}^{\star}) \tag{7}$$

where $p_\theta(\cdot|\boldsymbol{c},\boldsymbol{x})$ is an explicit likelihood, conventionally a Gaussian with mean and variance given by a neural network applied to $\boldsymbol{c},\boldsymbol{x}$. The CNP model is then trained by maximum likelihood, i.e. by minimizing the following conditional log probability

$$\mathcal{L}^{\text{CNP}}=-\mathbb{E}_F\left[\mathbb{E}_{\boldsymbol{x},\boldsymbol{y}}\left[\log q(\boldsymbol{y}^{\star}|(\boldsymbol{x}_i,\boldsymbol{y}_i)_{i=1}^{C},\boldsymbol{x}^{\star})\right]\right]. \tag{8}$$

Recall that the NP, unlike the CNP, includes an additional random variable $\boldsymbol{u}$. We can in fact view $\boldsymbol{u}$ as a finite dimensional approximation to $F$ in (2). In NPs, the random variable $\boldsymbol{u}$ is sampled from an approximate posterior $q\left(\boldsymbol{u}|(\boldsymbol{x}_i,\boldsymbol{y}_i)_{i=1}^{C}\right)$. The NP constructs the approximate posterior so that it is invariant to the order of the context, by using a sum pooling approach to aggregate the context. In order to learn this distribution, the NP introduces a modified training objective. Considering a context set $(\boldsymbol{x}_i,\boldsymbol{y}_i)_{i=1}^{C}$ and target set $(\boldsymbol{x}_i^{\star},\boldsymbol{y}_i^{\star})_{i=1}^{T}$, the NP training loss (Garnelo et al., 2018b) is

$$-\mathbb{E}\left[\mathbb{E}_{q\left(\boldsymbol{u}|(\boldsymbol{x}_i,\boldsymbol{y}_i)_{i=1}^{C},(\boldsymbol{x}_i^{\star},\boldsymbol{y}_i^{\star})_{i=1}^{T}\right)}\left[\sum_{i=1}^{T}\log q\left(\boldsymbol{y}_i^{\star}|\boldsymbol{x}_i^{\star},\boldsymbol{u}\right)+\log\frac{q\left(\boldsymbol{u}|(\boldsymbol{x}_i,\boldsymbol{y}_i)_{i=1}^{C}\right)}{q\left(\boldsymbol{u}|(\boldsymbol{x}_i,\boldsymbol{y}_i)_{i=1}^{C},(\boldsymbol{x}_i^{\star},\boldsymbol{y}_i^{\star})_{i=1}^{T}\right)}\right]\right] \tag{9}$$

where $q\left(\boldsymbol{y}^{\star}|\boldsymbol{x}^{\star},\boldsymbol{u}\right)$ is the explicit likelihood model, typically a Gaussian, as in the CNP. The outer expectation is with respect to the data $F,\boldsymbol{x},\boldsymbol{y}$.

**Attentive Neural Processes**    The ANP (Kim et al., 2019) introduced attention into the NP family in two different ways: *self-attention* applies to the context to create context-aware representations of each context pair $(\boldsymbol{x}_i,\boldsymbol{y}_i)$; *cross-attention* allows the ANP to attend to different components of the context depending on the target covariate $\boldsymbol{x}^{\star}$. These result in a representation $\hat{\boldsymbol{c}}(\boldsymbol{x}_{1:C},\boldsymbol{y}_{1:C},\boldsymbol{x}_i^{\star})$ that depends on $\boldsymbol{x}^{\star}$. As with the NP, the ANP can include a latent variable $\boldsymbol{u}$ to be sampled under a distribution that depends on $(\boldsymbol{x}_i,\boldsymbol{y}_i)_{i=1}^{C}$, self-attention can be used to generate the approximate

posterior for $\boldsymbol{u}$ in this case. The overall training loss for the ANP is

$$
\mathcal{L}^{\text{ANP}} = -\mathbb{E}_{F,\boldsymbol{x},\boldsymbol{y}} \left[ \mathbb{E}_{q\left(\boldsymbol{u}|(\boldsymbol{x}_i,\boldsymbol{y}_i)_{i=1}^C\right)} \left[ \sum_{i=1}^T \log q\left(\boldsymbol{y}_i^\star | \boldsymbol{x}_i^\star, \boldsymbol{u}, \hat{\boldsymbol{c}}(\boldsymbol{x}_{1:C}, \boldsymbol{y}_{1:C}, \boldsymbol{x}_i^\star)\right) \right] \right.
$$
$$
\left. -\text{KL}\left[ q\left(\boldsymbol{u}|(\boldsymbol{x}_i^\star, \boldsymbol{y}_i^\star)_{i=1}^T\right) \| q\left(\boldsymbol{u}|(\boldsymbol{x}_i,\boldsymbol{y}_i)_{i=1}^C\right) \right] \right]
$$

(10)

where $q\left(\boldsymbol{y}_i^\star | \boldsymbol{x}_i^\star, \boldsymbol{u}, \hat{\boldsymbol{c}}(\boldsymbol{x}_{1:C}, \boldsymbol{y}_{1:C}, \boldsymbol{x}_i^\star)\right)$ is the explicit likelihood model in this case. We refer to the ANP model without the latent $\boldsymbol{u}$ as the ACNP, for which the training loss is simply

$$
\mathcal{L}^{\text{ACNP}} = -\mathbb{E}_{F,\boldsymbol{x},\boldsymbol{y}} \left[ \sum_{i=1}^T \log q\left(\boldsymbol{y}_i^\star | \boldsymbol{x}_i^\star, \hat{\boldsymbol{c}}(\boldsymbol{x}_{1:C}, \boldsymbol{y}_{1:C}, \boldsymbol{x}_i^\star)\right) \right].
$$

(11)

**Transformer attention** The Image Transformer (Parmar et al., 2018) used an attention mechansim based on multi-head self-attention (Vaswani et al., 2017). To describe this attention using our notation, suppose that $\boldsymbol{r}_1, \ldots, \boldsymbol{r}_C$ are intermediate representations of pairs $(\boldsymbol{x}_1, \boldsymbol{y}_1), \ldots, (\boldsymbol{x}_C, \boldsymbol{y}_C)$. Then the $i$th representation $\boldsymbol{r}_i'$ in the next layer of representations is computed as follows. We apply a query linear operator $W_q$ to $\boldsymbol{r}_i$ and a key linear operator $W_k$ to $\boldsymbol{r}_j$ for $j = 1, \ldots, C$. We form a normalized set of weights

$$
w_{ij} = \frac{\exp\left(W_q \boldsymbol{r}_i \cdot W_k \boldsymbol{r}_j / \sqrt{d}\right)}{\sum_j \exp\left(W_q \boldsymbol{r}_i \cdot W_k \boldsymbol{r}_j / \sqrt{d}\right)}
$$

(12)

where $d$ is the dimension of $\boldsymbol{r}_i$. We then form a value as a weighted sum of existing representations, transformed with a value linear operator $W_v$ to give

$$
\tilde{\boldsymbol{r}}_i = \sum_j w_{ij} W_v \boldsymbol{r}_j.
$$

(13)

To convert $\tilde{\boldsymbol{r}}_i$ to $\boldsymbol{r}_i'$, we apply dropout, a residual connection (i.e. we add the original $\boldsymbol{r}_i$) and layer normalization (Ba et al., 2016). Then we apply a second fully connected layer with residual connection and layer norm to give $\boldsymbol{r}_i'$.

## Appendix C  Method details

### C.1  Downstream Tasks for Stochastic Processes

We provide some additional details on targeted and untargeted tasks. For a targeted task, we extend the stochastic process of Section 2 by introducing a second conditional distribution $p(\ell|F, \boldsymbol{x})$. We assume that the joint distribution over observations $\boldsymbol{y}_{1:C}$ and labels $\ell_{1:C}$ is given by

$$
p\left(\boldsymbol{y}_{1:C}, \ell_{1:C} | \boldsymbol{x}_{1:C}\right) = \int p(F) \prod_{i=1}^C p(\boldsymbol{y}_i|F, \boldsymbol{x}_i) p(\ell_i|F, \boldsymbol{x}) \, dF,
$$

(14)

implying that the predictive density of the label $\ell^\star$ at $\boldsymbol{x}^\star$ given the context $\{(\boldsymbol{x}_i, \boldsymbol{y}_i)_{i=1}^C\}$ is

$$
p\left(\ell^\star | \boldsymbol{x}^\star, (\boldsymbol{x}_i, \boldsymbol{y}_i)_{i=1}^C\right) = \frac{\int p(F) p(\ell^\star|F, \boldsymbol{x}^\star) \prod_{i=1}^C p(\boldsymbol{y}_i|F, \boldsymbol{x}_i) \, dF}{\int p(F) p(\ell^\star|F, \boldsymbol{x}^\star) \, dF}.
$$

(15)

In CRESP, we estimate this by forming a targeted representation $\hat{\boldsymbol{c}}$ of $(\boldsymbol{x}_i, \boldsymbol{y}_i)_{i=1}^C$ and $\boldsymbol{x}^\star$, and fitting a linear model $q(\ell|\hat{\boldsymbol{c}})$.

For untargeted tasks, there is one $\ell$ sampled along with the entire realization $F$ via a conditional distribution $p(\ell|F)$, giving the joint distribution

$$
p(\boldsymbol{y}_{1:C}, \ell | \boldsymbol{x}_{1:C}) = \int p(F) p(\ell|F) \prod_{i=1}^C p(\boldsymbol{y}_i|F, \boldsymbol{x}_i) \, dF.
$$

(16)

This means that we can predict $\ell$ using the context $\{(\boldsymbol{x}_i, \boldsymbol{y}_i)_{i=1}^C\}$ using the predictive distribution

$$
p\left(\ell^\star | (\boldsymbol{x}_i, \boldsymbol{y}_i)_{i=1}^C\right) = \frac{\int p(F) p(\ell^\star|F) \prod_{i=1}^C p(\boldsymbol{y}_i|F, \boldsymbol{x}_i) \, dF}{\int p(F) p(\ell^\star|F) \, dF}.
$$

(17)

In CRESP, we estimate this using a representation $\boldsymbol{c}$ of $(\boldsymbol{x}_i, \boldsymbol{y}_i)_{i=1}^C$; we fit a linear model $q(\ell|\boldsymbol{c})$.

# Appendix D  Experimental details

We provide below all necessary details to understand and reproduce the empirical results obtain in Sec. 5. Hyperparameters are summarized in Tab. 4. Models were implemented in PyTorch (Paszke et al., 2017). For downstream tasks we fit linear models with L-BFGS (Liu and Nocedal, 1989), we applied L2 regularization to the weights. Our code is available at `github.com/ae-foster/cresp`.

Table 4: Hyperparameters used for the different experiments.

| Parameter | Sinusoids | ShapeNet | Snooker |
|---|---|---|---|
| Covariate space $\mathcal{X}$ | $\mathbb{R}$ | $\mathbb{R}^{15}$ | $\mathbb{R}$ |
| Observation space $\mathcal{Y}$ | $\mathbb{R}$ | RBG 64x64 images | RBG 28x28 images |
| Dataset sizes | 17.6k/2.2k/2.2k | 26270/8756/8756 | 15k/3k/20k |
| Observation Net | Id | CNN | ResNet18 |
| Covariate Net | Id | Id | Id |
| Encoder Net | MLP | Gated | Gated |
| Decoder model | MLP | CNN | DCGAN |
| Attention | 2 transformer layers | 2 transformer layers | |
| Target network | | Gated | MLP |
| Training views | 10 | 3 | 5 |
| Test views | 20 | 10 | 9 |
| Representation dim | 512 | 512 | 512 |
| Projection dim | 128 | 128 | 128 |
| Training batch size | 256 | 512 | 256 |
| Training epochs | 200 | 10 | 200 |
| Optimizer | Adam | LARS | Adam |
| Scheduler | Cosine | Cosine + Ramp | Cosine + Ramp |
| Scheduler Ramp length | | 10 | 10 |
| Learning rate | 3e-4 | 2e-1 | 2e-3 |
| Momentum | 0.9 | 0.9 | 0.9 |
| Weight decay | 1e-6 | 1e-6 | 1e-6 |
| Temperature $\tau$ | 0.5 | 0.5 | 0.5 |
| Downstream L2 regularization | 1e-6 | 1e-3 | 1e-3 |

## D.1  CO$_2$ emissions

Experiments were conducted using a private infrastructure, which has an estimated carbon efficiency of 0.188 kgCO$_2$eq/kWh [2]. An estimated cumulative 1000 hours of computation was performed on hardware of type RTX 2080 Ti (TDP of 250W), or similar such as RTX 1080 Ti. Total emissions are estimated to be 47 kgCO$_2$eq. Estimations were conducted using the Machine Learning Impact calculator presented in Lacoste et al. (2019).

## D.2  Sinusoids dataset

**Data**  We sample unidimensional functions $F \sim p(F)$ such that $F(x) = \alpha \sin(2\pi/T \cdot x + \varphi)$ with random amplitude $\alpha \sim \mathcal{U}([0.5, 2.0])$, phase $\varphi \sim \mathcal{U}([0, \pi])$ and period $T = 8$. We assume a bimodal likelihood: $p(y|F, x) = 0.5\, \delta_{F(x)}(y) + 0.5\, \delta_{F(x)+\sigma}(y)$. Context points $x \in \mathcal{X}$ are uniformly sampled in $[-5, 5]$.

**Architectures**  Since both the covariate and observation variables are unidimensional, we do not preprocess them, i.e. $g_{\mathrm{cov}} = \mathrm{Id}$ and $g_{\mathrm{obs}} = \mathrm{Id}$. For the encoder–processing $g_{\mathrm{enc}}(g_{\mathrm{cov}}(\boldsymbol{x}), g_{\mathrm{obs}}(\boldsymbol{y}))$–we rely on an multilayer perceptron (MLP) with 3 hidden layer of 512 hidden units. For reconstructive methods (CNP and ACNP), the decoder is also parametrized by an MLP with 512 hidden units and 3 hidden layers.

---

[2]Average carbon intensity in March, April and June in the Great Britain. Source `https://electricityinfo.org/carbon-intensity-archive`.

### D.3 Shapenet dataset

**Data** We utilize the renderings of ShapeNet objects provided in 3D-R$^2$N$^2$ (Choy et al., 2016). These renderings are constructed from different orientations. We also apply a random crop to each image to simulate a random proximity to the object. Specifically, we choose a random area from $U(0.08, 1)$ and then a random crop of that area. This process is summarized by the PyTorch snippet

```
bounding_box = list(transforms.RandomResizedCrop.get_params(
    img, (0.08, 1), (1., 1.)
))
img = transforms.functional.resized_crop(
    img, *bounding_box, 64, Image.LANCZOS
)
```

This means that the covariate $x$ representing the view consists of the angles describing the orientation of the render, and the bounding box. We apply additional featurization to $x$ described in the next section. We also apply random colour distortion of strength $s$ as a noise process on the images $y$. Inspired by the colour distortion of Chen et al. (2020) we apply randomized brightness, contrast, saturation, hue and gamma adjustment (see our code for the exact implementation).

**Feature processing** We process the covariate $x$ as follows. For the azimuthal angle $\theta$, we use $\sin(n\theta), \cos(n\theta)$ for $n = 1, 2, 3$ and the original angle (7 features). We include the elevation and distance of the R$^2$N$^2$ render without additional features (2 features): in practice these vary little in this dataset. We include the bounding box mid-point and area as additional features, along with the four corners of the bounding box (6 features). All told, this gives a covariate of dimension 15. We finally apply normalization to the covariate so that each component has mean 0 and variance 1 over the entire dataset. To images $y$ we apply a linear rescaling that means each channel has mean 0 over the dataset.

**Learning set-up and downstream tasks** For unsupervized learning, we resample the view and distortion randomly each time an object is encountered. For learning on downstream tasks, we fix a dataset of covariates, observations and labels, and learn exclusively from this fixed dataset without resampling views, providing a more realistic semi-supervised test case. The labels are included in the dataset, but only utilized by our algorithm when we train downstream linear classifiers (except for the supervised baseline). The following 13 categories are represented in our dataset: display (1095), watercraft (1939), bench (1816), telephone (1052), cabinet (1572), sofa (3173), rifle (2373), loudspeaker (1618), airplane (4045), table (8509), chair (6778), car (7496), lamp (2318).

**Architectures** For the observation network, we use a CNN described by the following PyTorch snippet

```
nn.Sequential(
    nn.Conv2d(num_channels, ngf // 8, 3, stride=2, padding=1, bias=False),
    nn.BatchNorm2d(ngf // 8),
    nn.LeakyReLU(),
    nn.Conv2d(ngf // 8, ngf // 4, 3, stride=2, padding=1, bias=False),
    nn.BatchNorm2d(ngf // 4),
    nn.LeakyReLU(),
    nn.Conv2d(ngf // 4, ngf // 2, 3, stride=4, padding=1, bias=False),
    nn.BatchNorm2d(ngf // 2),
    nn.LeakyReLU(),
    nn.Conv2d(ngf // 2, ngf, 3, stride=4, padding=1),
    nn.BatchNorm2d(ngf),
    nn.LeakyReLU(),
)
```

and we set `ngf` $= 512$. For reconstructive methods (CNP and ACNP), we use a convolutional decoder of the following form

```
nn.Sequential(
```

```
      nn.UpsamplingNearest2d(scale_factor=2),
      nn.ConvTranspose2d(nz, ngf // 2, 2, stride=2, padding=0, bias=False),
      nn.BatchNorm2d(ngf // 2),
      nn.LeakyReLU(),
      nn.UpsamplingNearest2d(scale_factor=2),
      nn.ConvTranspose2d(ngf // 2, ngf // 4, 2, stride=2, padding=0, bias=False),
      nn.BatchNorm2d(ngf // 4),
      nn.LeakyReLU(),
      nn.ConvTranspose2d(ngf // 4, ngf // 8, 2, stride=2, padding=0, bias=False),
      nn.BatchNorm2d(ngf // 8),
      nn.LeakyReLU(),
      nn.ConvTranspose2d(ngf // 8, nc, 2, stride=2, padding=0),
)
```

where $\texttt{nz} = 512 + 15$, $\texttt{ngf} = 512$, $\texttt{nc} = 6$. Finally, we extract three means and three standard deviations from the output at each pixel location for three colour channels, applying a sigmoid to the means (to put them in the correct range for image data) and a softplus transform to the standard deviations.

The gated unit that we use is as follows

```
class Gated(nn.Module):

    def __init__(self, in_dim, representation_dim):
        super(Gated, self).__init__()
        self.fc1 = nn.Linear(in_dim, representation_dim)
        self.fc2 = nn.Linear(in_dim, representation_dim)
        self.activation = nn.Sigmoid()

    def forward(self, x):
        representation = self.fc1(x)
        multiplicative = self.activation(self.fc2(x))
        return multiplicative * representation
```

inspired by gated units that appear in Hochreiter and Schmidhuber (1997); Cho et al. (2014). The gated unit is utilized in two places: as the pair encoding (Sec. 3.2) that processes the covariate and observation features after concatenation, and as the target network for our targeted CRESP implementation on ShapeNet. We found that it slightly outperformed an MLP with a similar number of parameters.

### D.4 Snooker dataset

**Data** This synthetic dataset simulates a dynamical system with two objects evolving through time with constant velocities. Formally, let's consider two objects at positions $\mathbf{s}_i$ at time $t$. A free object moving at velocity $\boldsymbol{v}_i$ has position $\mathbf{s}_i(t) = \mathbf{s}_i(0) + \boldsymbol{v}_i t$. We now consider both objects constrained so that $0 \leq \mathbf{s}_i \leq 1$ and assume that collisions with the boundaries result in a perfect reflection. The position of the particle can be expressed by the following formula

$$\tilde{s}_i(t) = s_i(0) + v_i t, \tag{18}$$

$$s_i(t) = (\lfloor \tilde{s}_i(t) \rfloor \mod 2)(1 - \tilde{s}_i(t) + \lfloor \tilde{s}_i(t) \rfloor) + (1 - \lfloor \tilde{s}_i(t) \rfloor \mod 2)(\tilde{s}_i(t) - \lfloor \tilde{s}_i(t) \rfloor) \tag{19}$$

for $i = 1, 2$.

We then assume that we only have access to a 2D image $\boldsymbol{y}$ of the state at time $\boldsymbol{x} = t$ for a given realization $F$. We sample realizations $F \sim p(F)$ such that $\mathbf{s}_i(0) \sim \mathcal{U}([0,1]^2)$, and $\boldsymbol{v}_i = v_0 \boldsymbol{\alpha}$ with $\boldsymbol{\alpha} \sim \mathcal{U}(\mathbb{S}^1)$ and $v_0 = 0.4$. The objects are assumed to be non-interacting discs of radius $0.15$.

The downstream task is to predict whether the two objects are overloading at a given time, i.e. $\mathbb{E}_{p(\ell|F,\boldsymbol{x}^\star=t)}[\ell]$ with $\ell = 1$ if there is an overlap. The objects position can be expressed at any time in closed-form (cf Eq. (18)), yet it is quite challenging to predict the 2D image at a specific time given a collection of snapshots.

**Architectures**  For the observation network, we use a CNN described by the following PyTorch snippet

```
nn.Sequential(
    nn.Conv2d(nc, ngf, kernel_size=2, stride=2, bias=False),
    nn.BatchNorm2d(ngf),
    nn.ReLU(True),
    nn.Conv2d(ngf, 2 * ngf, kernel_size=2, stride=2, bias=False),
    nn.BatchNorm2d(2 * ngf),
    nn.ReLU(True),
    nn.Conv2d(2 * ngf, 4 * ngf, kernel_size=2, stride=2, bias=False),
    nn.BatchNorm2d(4 * ngf),
    nn.ReLU(True),
    nn.Conv2d(4 * ngf, nz, kernel_size=2, stride=2),
)
```

where $\texttt{ngf} = 64$ and $\texttt{nc} = 3$. For reconstructive methods (CNP and ACNP), we use a convolutional decoder inspired by DCGAN (Radford et al., 2016), of the form

```
nn.Sequential(
    nn.ConvTranspose2d(nz, ngf * 4, 4, 1, 0, bias=False),
    nn.BatchNorm2d(ngf * 4),
    nn.ReLU(True),
    nn.ConvTranspose2d(ngf * 4, ngf * 2, 3, 2, 1, bias=False),
    nn.BatchNorm2d(ngf * 2),
    nn.ReLU(True),
    nn.ConvTranspose2d(ngf * 2, ngf, 4, 2, 1, bias=False),
    nn.BatchNorm2d(ngf),
    nn.ReLU(True),
    nn.ConvTranspose2d(ngf, 2 * nc, 4, 2, 1),
)
```

where $\texttt{nz} = 512 + 1$, $\texttt{ngf} = 64$ and $\texttt{nc} = 2 * 3$. Similarly to Appendix D.3, we extract three means and three standard deviations from the output at each pixel location.

For the encoder–processing $g_{\mathrm{enc}}(g_{\mathrm{cov}}(\boldsymbol{x}), g_{\mathrm{obs}}(\boldsymbol{y}))$–we rely on the gated architecture described above in Appendix D.3. For the target network $h$, which outputs the predictive representation $\hat{\boldsymbol{c}} = h([\boldsymbol{x}^{\star}, \boldsymbol{c}])$, we rely on an MLP with 3 hidden layers of 512 units each.