# OpenReview forum: "On Contrastive Representations of Stochastic Processes"
_NeurIPS.cc/2021/Conference — NeurIPS 2021 Poster_

### Official Review · Reviewer_z4gr · 2021-07-13

**Rating:** 5
**Confidence:** 3

**Summary:**

My understanding of the paper is that authors propose to introduce contrastive methods (Van den Oord 2019, Chen 2020) to improve the learning of stochastic processes with NN architectures. Particularly, they identify the main points where the Neural Process (NP+CNP) family of models fails, e.g. due to high-dimensionality or complex likelihood densities. The advantage of using these contrastive representations for stochastic processes is that it makes the task likelihood-free, performing better when the aforementioned problems appear.

**Limitations And Societal Impact:**

Limitations and societal impact are correctly addressed. Authors indicate that the method is not able to directly predict on the observation space as one could desire. Moreover, they indicate the advantages of using the model for better understanding of 3D scenes, what might imply some risks as well.

**Main Review:**

**Originality and significance:**

It can be clearly seen that the main 3 references for this work are:

- Garnelo (2018) / conditional neural processes
- Van den Oord (2019) / contrastive representations
- Chen (2020) / contrastive learning

The contribution is also clear to me: to introduce the concept of contrastive learning to improve the learning mechanism used in NP+CNPs. Despite the work is a mix of the previous references, I see that it is definitely interesting and it might have a future impact. However, I think that the ideas are not that well developed through the paper (e.g. the likelihood-free inference and its importance for the CNP models)… particularly due to a lack of clarity. That is the main flaw, in my opinion.

- The training method in Eq. (6) is super similar to Chen (2020) as authors recognise. What does attention/aggregation steps imply in this particular scenario?

**Quality and clarity:**

- I find the choice of notation a bit counterintuitive, mainly for those familiar with stochastic processes or just the functional expression y=f(x). For instance, the use \xi, x for the context/dataset, where \xi are the covariates and x the output of the function is difficult to follow, at least for me. In the original NP and CNP papers, the usual notation {x,y} were used. I do not see the reason why this is not preserved. (Sometimes the notation is changed due to other variables are preferred to have other names).

- The paper is well motivated on stochastic processes, that’s great, and it is also well written until L110 approx. Later things are more difficult to follow. In particular I see that the loss of contrastive learning, in Eq. (3) and so on and the use of the likelihood ratio is not clearly explained. (See Section 2.1 of Chen (2020) as an example.) This is important, since contrastive learning is the key contribution, and the reader should understand very well what authors are introducing in the neural/stochastic process mechanism.

- Targeted tasks/untargeted tasks are not motivated, so it is difficult to figure out an example. Until L137 it is +/- clear what the elements of the model are.. context data: {\xi, x}, representation c, code z… but where does y come from?

- From a technical point of view, the paper is relevant. However, I am not fully convinced by the experiments. For example, the section 5.1. with the sinusoids and the multimodality problem: using such a large architecture for learning the parameters {\alpha, \phi} seems a bit problematic. Could be this problem easily solved by a simpler stochastic process? i.e. GP or similar? To me, is a bit weird to see that the CNP model fails here, particularly, if one looks to the original paper. This is because of the multivariate and multimodal likelihood problem, I guess.

**Questions and other comments:**

- What does a “negative sample” z_1, z_2, … (L175) mean?
- L185: more more
- L174: how is this projection done?

**References:**

Van der Oord et al. 2019, “Representation Learning with Contrastive Predictive Coding”. ArXiv
T. Chen et al. 2020, "A Simple Framework for Contrastive Learning of Visual Representations" ICML 2020
M. Garnelo et al. 2018, “Conditional Neural Processes”, ICML 2018


**Time Spent Reviewing:**

3,5

---

> ### Author Response · Authors · 2021-08-10
> **Reply to Reviewer z4gr**
>
> Thank you for your detailed feedback on our work.
> &nbsp;
>
> > The training method in Eq. (6) is super similar to Chen (2020) as authors recognise. What does attention/aggregation steps imply in this particular scenario?
>
> SimCLR creates representations of single images. CReSP creates representations of samples from stochastic processes. For instance, CReSP takes multiple photos of a 3D object, plus the angles from which these images were captured, and creates *one* representation for the 3D object as a whole. SimCLR would give you one representation per image. As you point out, the added complexity is how to combine the different inputs, which is where the attention and aggregation comes in. So, for the 3D object example, the attention mechanism could learn to place greater weight on certain viewpoints (side on, say) compared to other viewpoints (view from underneath), upweight sharper images, etc. CReSP also applies to other stochastic processes, such as time-evolving processes. Here the attention has to choose which time-separated events to focus on. To summarise, SimCLR only handles one input whereas CReSP works with a set of pairs coming from a stochastic process.
> Our paper is more similar to the recent FCLR paper (ICML 2021), but with some important differences that we spell out in lines 75-77 and 241-245.
> &nbsp;
>
> > I find the choice of notation a bit counterintuitive
>
> We initially chose to keep the variable y to refer to the downstream task label, but we agree that it can add unnecessary confusion for readers and will revert to the $y=f(x)$ notation, and use a different letter such as $\ell$ for the downstream label.
> &nbsp;
>
> > the loss of contrastive learning, in Eq. (3) and so on and the use of the likelihood ratio is not clearly explained
>
> We will enhance this paragraph to improve clarity. The likelihood ratio is introduced in Eq. (4) to answer the question “how do we do maximum likelihood learning as in a CNP (as in lines 100-103) without an explicit likelihood model?” Our idea is to try and maximise the likelihood ratio in Eq. (4), which can be estimated by the contrastive loss in Eq (5). The point is that we are solving the *same learning problem as in the CNP*, but we no longer have to have an explicit likelihood.
> &nbsp;
>
>
> > Targeted tasks/untargeted tasks are not motivated
>
> We aimed to first introduce a rigorous and abstract framework for our method and later on to motivate the distinction between targeted and untargeted tasks from a practical perspective. Untargeted tasks are about predicting labels for the entire process realisation: $p(y|F)$, as opposed to targeted tasks which are query dependent $p(y|\xi^\star,F)$. Examples of untargeted tasks include function parameter identification, object classification or regression, while targeted tasks include learning state representation for reinforcement learning or predicting hazardous situations for self-driving cars.
> &nbsp;
>
> > where does y come from?
>
> This is discussed in Section 3.3: we take a semi-supervised approach and assume access to a small amount of labelled data. However, we thank you for the suggestion and agree that delaying this until Sec 3.3 makes things confusing- we plan to bring some of this information earlier so that it is easier to follow.
> &nbsp;
>
> > Could be this problem easily solved by a simpler stochastic process? i.e. GP or similar?
>
> With the right prior knowledge, a simple parametric stochastic process would indeed be able to fit the data perfectly. The motivation for neural approaches like CReSP is that no such strong prior knowledge is required to be able to fit the data well, which makes our models applicable to a broad range of problems. In this task, we indeed show that the CNP is able to accurately recover the parameters when the data is (close to) unimodal, yet that it fails when the multimodality is too strong.
> &nbsp;
>
> > What does a “negative sample” z_1, z_2, … (L175) mean?
>
> ‘Negative samples’ line 175 means samples that are coming from other realisations of the stochastic process, so in practice representations obtained from the other elements of the batch. For example, in the 3D object example, if z* is a representation of the target view of an aeroplane (say, a side on view), then the negative samples are representations of random views of random other objects such as a top view of a car. The term ‘negative sample’ comes from the contrastive learning literature as the InfoNCE loss can be seen as the categorical cross-entropy of classifying the ‘positive’ samples correctly (Oord et al., 2018).  &nbsp;
>
>
> L185: thanks, we’ll correct.
> &nbsp;
>
> > L174: how is this projection done?
>
>  We use a shallow MLP.

---

> > ### Comment · Reviewer_z4gr · 2021-09-02
> > **After rebuttal**
> >
> > Thanks to the authors for their response and clarification of the main questions that I included in my review. I still think that the introduction of contrastive learning into the neural process mechanism is interesting, particularly on the point of making tasks likelihood-free. My main concerns about the clarity of the work have been +/- addressed in the response, but I still see that the presentation of the methodology could be better presented. As I also said in my review, the experiments do not fully convince me of the performance of the methodology. That's the main reason why I will keep my score as it is now. If accepted, would be nice to see the methodology and notation somehow improved for having a higher impact in the community; if not, I will be looking forward to see a revised version of the work in future conferences.

---

### Official Review · Reviewer_oGtj · 2021-07-13

**Rating:** 6
**Confidence:** 4

**Summary:**

This paper proposes a meta-learning model that represents stochastic processes through a contrastive representation. The authors mainly addressed the unified framework which could handle the two types of tasks; 1) given a set of data and target covariate (they called it the targeted task) and 2) estimating the label for the set of data (called untargeted task). To do that, they designed the specific encoder for the targeted task and by applying a self-attention mechanism on their encoder, they improved the performance for the tasks. They validated their model on interesting tasks like bi-modal sinusoidal regression and semi-supervised learning. I think that the architectural novelty of their model is the unified design for targeted/untargeted tasks and self-attention layers. However, I don't think their contribution is not limited to them, they validated/analyzed their model and other meta-learning models trained through the contrastive losses (e.g., FCLR and MetaCDE) for a variety of interesting tasks, which is one of the strong points on this paper I thought.

**Limitations And Societal Impact:**

Their model has been validated for quite simple labels like parameters of sinusoids or one of 13 class labels. However, those models with contrastive loss are more powerful for visually complicated tasks like 3D map world model for agent. For those tasks, the representations $c$ are required to contain enough knowledge for environment. I am not sure that CReSP can cover those complex tasks also.

**Main Review:**

When I first review this paper, I felt it is quite similar to FCLR. I focused to find the difference between FCLR and CReSP, and as I understand, the main differences are 1) targeted encoder for the targeted task and 2) self-attention on the encoder. The authors explained it as CReSP is the unified framework for meta-learning with contrastive loss. It could be seen as incremental work, but they validated their model and previous works on several interesting tasks including targeted and untargeted tasks. They pretty well analyzed the results (e.g., for the ShapeNet task, why the performances of CReSP and FCLR decrease when the number of given views increases), and their model outperforms previous models, so I think that we cannot say easily it is just incremental work. Overall writing is good to read but the notation for covariate $\xi$, observation $x$ and label $y$ caused some confusion for me, because usually, NPs papers use $x$ as the covariate and $y$ for observation. If one of your contributions is the unified framework for meta-learning model with contrastive loss, then using the below storyline as main could make this paper more understandable.

Using contrastive loss for meta-learning is very beneficial -> FCLR was proposed ->  but it cannot cover targeted task well, so we proposed unified framework including the FCLR

Questions / comments
- Line 103-4, you mentioned NP model is the model with additional latent variable on top of CNP, but it is not correct. CNP samples the functional latent when decoding with each target query, which causes inconsistant uncertainty, so the authors updated their CNP to sample the functional latent before decoding to resolve the inconsistant uncertainty problem. GQN that is origin of CNP (CNP is generalized version of GQN) is also inconsistant for uncertain knowledge, so Consistant GQN (Ananya et al., 2019) was proposed.
- Line 203-7, what is the meaning of equation in line 204? self-attention? From my perspective, the equation causes not necessary confusion. Why did you use Image Transformer, not general Transformer? The performance when using Image Transformer was better?
- For Sinusoids task, how did you use ACNP for untargeted task? Did you summarize the representations for the context set?
- In Fig. 3, second image in second row is not distorted.


**Time Spent Reviewing:**

9 Hours

---

> ### Author Response · Authors · 2021-08-10
> **Reply to Reviewer oGtj**
>
> Thank you for a thorough and helpful review.
> We are very pleased that you felt we rigorously evaluated our method against competitive baselines on a number of interesting tasks. We strongly agree with you that the paper would gain from updating the storyline so as to highlight the benefit of contrastive learning for meta-learning, which our model does by building on and including FCLR as a special case.
> &nbsp;
>
> > the notation ... caused some confusion for me
>
> Regarding clarity of the notation, we initially chose to keep the variable y to refer to the downstream task label, but we agree that it can add unnecessary confusion for readers. We will revert to the $y=f(x)$ notation, adopting another letter $\ell$ for the downstream label.
> &nbsp;
>
> > inconsistant uncertainty problem
>
> Indeed neither the NP or CNP are formally consistent, hence are not technically stochastic processes. We thank you for the suggestion and will update Section 2 so as to reflect this and avoid any confusion.
> &nbsp;
>
> > what is the meaning of equation in line 204?
>
> The meaning of the equation with $\text{attn}(\{g_{enc}(\xi_i,x_i)\}_{i=1}^C) $ is to indicate that self-attention is performed. We agree that it may cause unnecessary confusion and will remove it.
> &nbsp;
>
> > Why did you use Image Transformer, not general Transformer?
>
> We tried multiple attention mechanisms, such as ‘multiplicative’, ‘dot product’, ‘euclidean’ and ‘additive’ (the original parameterization [1]), but quickly found out that image transformer attention [2] was significantly outperforming others. We use the term ‘transformer attention’ to specifically refer to the attention mechanism of the image transformer paper. We will make this more explicit in the paper.
> &nbsp;
>
> > For Sinusoids task, how did you use ACNP for untargeted task?
>
> Regarding the sinusoids task and ACNP, as it is an untargeted task, we followed the first part of the architecture proposed in [3], which is a self-attention layer on the (query, observation) pairs and then applied a mean aggregation layer to obtain the representation.
>
> Thanks for picking up the error in Fig 3, we will ensure that it is corrected.
> &nbsp;
>
> > the representations z are required to contain enough knowledge for environment. I am not sure that CReSP can cover those complex tasks also.
>
> This is definitely an interesting point.
> Although 3D world tasks are challenging, we showed that on the dynamical process collision detection task, the learnt representations contain enough information about the environment to accurately detect whether a collision will occur at a specific time or not.
> One of the key differences between real environments and toy environments is the presence of high-dimensional observations with naturalistic noise (blur, etc). We believe this is a case where the contrastive approach can bring an edge- as we verified on the ShapeNet experiment, naturalistic noise significantly damages explicit likelihood methods, but CReSP continues to perform well with more distortion.
> &nbsp;
>
>  [1] Luong, Minh-Thang, Hieu Pham, and Christopher D. Manning, Effective approaches to attention-based neural machine translation, 2015
>  [2] Parmar, Niki, et al., Image transformer, 2018
>  [3] Hyunjik Kim, Andriy Mnih, Jonathan Schwarz, Marta Garnelo, Ali Eslami, Dan Rosenbaum, Oriol Vinyals, Yee Whye Teh, Attentive Neural Processes, 2019

---

### Official Review · Reviewer_S13o · 2021-07-15

**Rating:** 4
**Confidence:** 4

**Summary:**

This paper proposed a framework for learning contrastive representation of stochastic processes (CRESP) and designed two variants of models to tackle targeted and untargeted downstream tasks (e.g. label prediction) in a semi-supervised setting.  The models were evaluated on downstream predictive tasks of three stochastic processes and outperform previous contrastive learning methods.

**Limitations And Societal Impact:**

A limitation has already been mentioned in this paper, the models proposed in this paper aim to learn the representations from stochastic processes and further be used in the downstream tasks. It cannot predict observation from stochastic processes. A net for reconstruction from c to x may be helpful to improve the learned representation.


**Main Review:**

The structure of this paper is clear and easy to read. The research in this paper is very important for the comparative learning of stochastic processes. Overall, it is an ok paper but not enough for acceptance.

This core idea of models proposed in this paper is very similar as the idea in paper  SimCLR (Chen et al. 2020). The novelty of this paper is low.

These figures lack some necessary variables. For example Figure 1 should also show \hat{z}, z*, and z’.

**Time Spent Reviewing:**

3

---

> ### Author Response · Authors · 2021-08-10
> **Reply to Reviewer S13o**
>
> Thank you for your thoughtful review, we are pleased that you found our work clear and easy to follow.
> &nbsp;
>
> > This core idea of models proposed in this paper is very similar as the idea in paper SimCLR
>
> This is not correct. SimCLR learns representations of individual images, whereas CReSP learns representations of contexts sampled from stochastic processes. These models are therefore tackling different tasks. For example, CReSP could be trained to create *one* representation from multiple images of a 3D object taken from different viewpoints, SimCLR is trained to create one representation per image.
> Our paper is more similar to the recent FCLR paper (ICML 2021), but with some important differences that we spell out in lines 75-77 and 241-245.
> &nbsp;
>
>
> > Figure 1 should also show \hat{z}, z*, and z’
>
> The more detailed figure you suggest can be found on page 3 of the supplement.

---

### Official Review · Reviewer_jJb1 · 2021-07-16

**Rating:** 5
**Confidence:** 3

**Summary:**

The authors propose a method for representation of an observational context by optimization of a contrastive learning objective. In contrast to (conditional) neural process modeling, this avoids specification of a likelihood, which may be difficult and ultimately unnecessary in tasks that only require prediction of a downstream label. Experimental results illustrate the advantages of this method over related neural process models on such tasks.

**Limitations And Societal Impact:**

The authors acknowledge some limitations in terms of their model's inability to reconstruct data in the observation space, in contrast to other neural process methods. Per my concern above, I would like to understand whether the limitations are in fact more broad from the standpoint of modeling with stochastic processes - in particular, whether the method does not actually provide tools for quanitfying posterior uncertainty over contexts or downstream predictions.


**Main Review:**

This paper suggests a contrastive learning alternative to likelihood-based methods for training neural processes, which are themselves a recent attempt to overcome the computational issues known to hinder inference in stochastic (Gaussian) process models. In tasks that depend on some downstream property of the prediction, rather than the prediction itself, reconstruction is unnecessary and likelihood-free methods may be used for learning the context representation that features in the neural approximation to the posterior. Here, the authors propose a contrastive learning objective based on InfoNCE.

The paper is clear and detailed, including careful documentation of experimental data, preprocessing, and hyperparameters in the Appendix. The synthetic experiment offers a clear illustration of the relative advantage over (conditional) neural processes when the likelihood is misspecified. Experiments on image and dynamical process data show that the framework can be tailored to address a broader range of tasks (targeted and untargeted, i.e. conditional or not on the prediction at a given index) than FCLR and MetaCDE, two recent proposals for contrastive learning.

My main concern is that the stochastic process perspective taken in the paper seems only tenuously related to the actual training and prediction schemes detailed in Sec 3.1-3.3. In practice, the method learns an embedding for observational contexts via minimization of a contrastive loss, and this embedding is subsequently applied deterministically at test time to generate a fixed-dimensional feature representation for a linear model of the downstream task. The only notion of a predictive distribution that remains is from this linear modeling step (Appendix D, p. 2); there is no posterior uncertainty over contexts. These issues undermine the framing of this method and its comparison to (conditional) neural processes, which intentionally retain an emphasis on quantifying uncertainty.


**Time Spent Reviewing:**

5

---

> ### Author Response · Authors · 2021-08-10
> **Reply to reviewer jJb1**
>
> Thank you for your review. We are really pleased that you appreciate the clarity of the paper, and recognise the value of likelihood-free learning in a representation learning setting.
> &nbsp;
>
>
> > the stochastic process perspective taken in the paper seems only tenuously related to the actual training and prediction schemes
>
> We would first like to emphasise that the stochastic process is an *assumption about the data generating distribution*. It is a way to formally specify the kinds of data that our algorithm operates on, and rigorously justifies our encoder architecture, particularly the permutation invariance approach that we take to aggregation.
>
> As such, we do not see that using a deterministic embedding of context data that is sampled from a stochastic process is invalid. This general approach has been taken by numerous other works, including CNP and FCLR.
>
> Similarly to the NP family, the predictive distribution over the label $p(y^\star|\xi^\star,(\xi,x_i)_{i=1}^C)$ incorporates predictive uncertainty via a softmax parameterization (in the classification setting) or a predictive variance (in the regression setting).
>
> We very much agree that further discussion on this point would add to the paper, and we will be sure to add this to a revised version of the paper.
> &nbsp;
>
> > there is no posterior uncertainty over contexts
>
> This is correct, we do not explicitly place a posterior on contexts or context representations. We agree with the reviewer that additionally learning stochastic embeddings of contexts is an interesting idea worth pursuing as future work. There is no trivial nor unique way to tackle this. One approach would be to utilise recent literature that extends the InfoNCE loss to deal with distributions [1]. The primary benefit of this would be to produce correlated predictions at two or more covariates. This would form an extension to the current submission. We will make sure to add a discussion to the future work section of the paper about this.
> &nbsp;
>
>
> > These issues undermine the framing of this method and its comparison to (conditional) neural processes, which intentionally retain an emphasis on quantifying uncertainty.
>
> To reiterate, CReSP has the same capacity to quantify uncertainty as other methods with deterministic embeddings (CNP, FCLR).
>
> [1] A Simple Framework for Uncertainty in Contrastive Learning, 2020, Mike Wu, Noah Goodman.

---

> > ### Comment · Area_Chair_CXp6 · 2021-08-28
> > **does the rebuttal satisfy?**
> >
> > Hi reviewer jJb1,
> >
> > In the rebuttal, the authors refute your main concern with the work: do you agree?
> >
> > Thanks.

---

> > > ### Comment · Reviewer_jJb1 · 2021-08-29
> > > **on this point, yes**
> > >
> > > I appreciate the authors’ reply and acknowledge their point. The authors’ terminology and approach does appear in line with recent work in this area (e.g. FCLR). Some clarification may help readers who might otherwise expect the method to work with some probabilistic representation of the context. I think that addressing this case would constitute a substantial methodological contribution and clear point of contrast to the related works discussed in the manuscript.

---

### Author Response · Authors · 2021-08-10
**Overall response to all reviewers**

We thank the reviewers for their time, helpful feedback, and thoughtful advice. We are pleased that overall, reviewers praised the clarity [jJb1, S13o, oGtj], rigour and interesting range of applications of the work [jJb1, oGtj]. We are encouraged that some reviewers acknowledged novelty [oGtj, z4gr] and appreciated our work as a principled contribution in the development of machine learning methods for learning function representations.

Several reviewers asked how our work differs from SimCLR. We would like to stress that although both SimCLR and our work use the contrastive infoNCE loss (which goes back to Oord et al., 2018 and earlier work), they are fundamentally solving different tasks. SimCLR creates representations of single images, whereas CReSP creates representations of samples from stochastic processes. CReSP must therefore use different architectures to SimCLR, incorporating attention and aggregation layers, which are completely absent in SimCLR. Our paper is more similar to the recent FCLR paper (ICML 2021), but with some important differences that we spell out in lines 75-77 and 241-245.

Several reviewers suggested changing the $\xi,x$ notation to the more familiar $x, y$ notation for stochastic processes. We agree and will make the change, using another letter such as $\ell$ for the downstream task label.

We have responded to reviewers' individual concerns in separate comments.

---

### Decision · Program_Chairs · 2021-09-27

**Decision:**

Accept (Poster)

**Comment:**

One of the reviews lacks thoroughness - I'm happy to disregard this.

The work proposes meta learning through neural processes and contrastive learning. There is some disagreement about the level of contribution here, with several reviewers pointing out that the FCLR method also contributes contrastive learning, and another reviewer pointing out that this is not so incremental because of extensions in several directions.

One reviewer raises a strong technical objection, that the "stochastic process perspective taken in the paper seems only tenuously related to the actual training and prediction schemes". The authors refute - "using a deterministic embedding of context data that is sampled from a stochastic process is (not) invalid". the authors agree to amend the work to improve clarity, and the reviewer conceded that clarifying this point in the manuscript would make for a strong contribution.

One reviewer raised issues with the experiments - that they do illuminate the method or support the proposed ideas. Yet I'm inclined to agree with another reviewer who found the detailed analysis, descriptions, and hyperparameters in the supplementary material. I also approve of the experiment to show limitations of the method in learning the strongly multimodal case.

Overall, I think the authors have enough feedback from the reviewers to make some tweaks to some of the explanations in the paper, making this a contribution to the NeurIPS community.